



# Bridging the scale gap: Obtaining high-resolution stochastic simulations of gridded daily precipitation in a future climate

Qifen Yuan[1,2], Thordis L. Thorarinsdottir[3], Stein Beldring[1], Wai Kwok Wong[1], and Chong-Yu Xu[2]

[1]Norwegian Water Resources and Energy Directorate, Oslo, Norway
[2]University of Oslo, Oslo, Norway
[3]Norwegian Computing Center, Oslo, Norway

**Correspondence:** Qifen Yuan (qiyu@nve.no)

**Abstract.** Climate change impact assessment related to floods, infrastructure networks and water resources management applications requires realistic simulations of high-resolution gridded precipitation series under a future climate. This paper proposes to produce such simulations by combining a weather generator for high-resolution gridded daily precipitation, trained on historical observation-based gridded data product, with coarser scale climate change information obtained using a regional climate

model. The climate change information can be added to various components of the weather generator, related to both the probability of precipitation as well as the amount of precipitation on wet days. The information is added in a transparent manner, allowing for an assessment of the plausibility of the added information. In a case study of nine hydrological catchments in central Norway with the study areas covering 1000-5500 km$^2$, daily simulations are obtained on a 1 km grid for a period of 19 years. The method yields simulations with realistic temporal and spatial structures and outperforms empirical quantile delta

mapping in terms of marginal performance.

## 1   Introduction

The rate of projected future warming in Northern Europe is amongst the highest in the world, driven to a large extent by the strong feedback involving snow and ice as the climate warms (Collins et al., 2013). As a consequence, the hydrological cycle intensifies (Bengtsson, 2010) leading to more precipitation as well as more intense extreme events (e.g. Vautard et al.,

2014). The projected changes in precipitation amounts, snowpack and snow cover will considerably impact surface hydrology through, for example, changed surface runoff as well as timing and amplitude of the spring flood (e.g. Von Storch et al., 2015). In order to study these effects, impact models optimally require inputs that reliably represent precipitation occurrence and intensity at a high spatial resolution, spatial and temporal variability, as well as physical consistency for different regions and seasons (Maraun et al., 2010).

Coupled atmosphere-ocean general circulation models (GCMs) remain our main source of information for projections of future climate. However, these have spatial resolutions that are too coarse for assessing the often localized impacts of changing precipitation patterns. Regional climate models (RCMs) at a spatial resolution of 10-15 km (e.g. Jacob et al., 2014) are able to explicitly resolve mesoscale atmospheric processes and add valuable information for precipitation modeling over a region, with the newest model generations at an even higher resolution and able to include explicit deep convection (Lind et al., 2020).





To obtain reference results for current climate, impact models are commonly applied to high-resolution historical data products such as the Nordic Gridded Climate Dataset (NGCD, https://surfobs.climate.copernicus.eu/dataaccess/access_ngcd.php) which provides historical estimates of precipitation and temperature in Northern Europe at a 1 km spatial resolution. Such data products come with their own inherent biases which can be difficult to correct due to lack of data. For an accurate assessment of climate impact, one goal is thus to generate high-resolution realizations of future climate which properties differ from those

describing the reference climate only in terms of the expected climate change between the two time periods. For comparable future projections, RCM simulations need a further downscaling step, and systematic biases as well as incompatibilities between the two spatial scales should be removed. It has further been argued that downscaling should be stochastic in nature and able to generate sub-grid spatial variability (Maraun et al., 2017). Recently proposed stochastic downscaling methods have proven skillful in modeling the small-scale variability of precipitation occurrence and intensity across sets of point locations

(Wong et al., 2014; Volosciuk et al., 2017).

       This paper proposes a two-stage weather generator (WG) approach to generate high-dimensional simulations of future climate on a fine scale grid. Specifically, a stochastic model describing a high-resolution data product in a reference period is combined with climate change projections based on a lower resolution RCM. Weather generators are commonly used to generate spatially and temporally correlated fields of daily precipitation, with the early work of Wilks (1998) paving the way

for many current approaches. Chandler and Wheater (2002) illustrate the use of a generalized linear model (GLM) to describe daily precipitation series at individual sites, using a logistic regression model for the occurrence and a gamma model for the amounts. More recently, Kleiber et al. (2012) propose an approach relying on two latent Gaussian random fields to generate spatially correlated occurrence and intensity, with spatial heterogeneity described through both spatially varying covariates and regression parameters. Serinaldi and Kilsby (2014) propose a more computationally efficient approach, where a single latent

Gaussian random field is used to describe the spatial correlation in both precipitation occurrence and intensity.

       With applications related to hydrological impacts in mind, we consider a case study of nine different catchments in central Norway. The simulation of daily fine-scale precipitation for a catchment requires daily simulations of spatially correlated random fields on a high-resolution grid with roughly 1000-5500 grid cells, depending on the size of the catchment. As the catchments are located in different climatic zones, the stochastic model is estimated independently for each catchment. Spatial

heterogeneity within a catchment is introduced via spatially varying covariates for both the occurrence and the intensity models, where the covariate contribution to the precipitation intensity may vary smoothly in space. Additionally, temporal aspects are modeled with seasonal effects and linear trends in the marginal distributions, as well as an autoregressive component in the residual process. Climate change information from an RCM output may be added in a transparent manner by updating each component of the weather generator based on estimated climate change in the corresponding component at the coarser RCM

scale. Yuan et al. (2019) propose a similar model for obtaining high-resolution daily mean temperature projections.

       The remainder of the paper is organized as follows. Section 2 introduces the datasets and the study area. Details of the two-stage WG approach are given in Section 3, together with a description of a reference method based on empirical quantile delta mapping as well as the evaluation methods used to compare the two approaches. The models are estimated based on data from the period 1957-1986 and the estimates are used to simulate data for the period 1987-2005. The results of this analysis





and comparison of the various approaches is given in Section 4. The paper then concludes with a brief summary and discussion in Section 5.

## 2 Data and study area

We apply our methodology to daily precipitation simulations from two RCMs from the EURO-CORDEX-11 ensemble. One (referred to as RCM1 in the following) combines the COSMO Climate Limited area Model (CCLM) from the Potsdam Institute

for Climate Research (Rockel et al., 2008) with boundary conditions from the CNRM-CM5 Earth system model developed by the French National Centre for Meteorological Research (Voldoire et al., 2013), whereas the other (referred to as RCM2) combines the CCLM model with boundary conditions from the MPI Earth system model developed by the Max Planck Institute for Meteorology (Giorgetta et al., 2013). The RCM simulations are conducted over Europe at a spatial resolution of 0.11 degrees or about 12 km (Jacob et al., 2014). In the historical period up to 2005 the outputs are simulated based on recorded emissions

and are thus comparable to observed climate.

For observational reference data, we use the seNorge gridded data product version 2018 produced by the Norwegian Meteorological Institute (Lussana et al., 2019), as subset of the NGCD for Norway. The data result from a multi-scale spatial interpolation of measurements from 500-700 surface weather observation stations for the period 1957 to present. The data have a daily temporal resolution and a spatial resolution of 1 km over an area covering the Norwegian mainland and an adjacent strip

along the Norwegian border. Compared with previous versions of the data product (i.e., Lussana et al., 2018), seNorge version 2018 adjusts the measurements for wind-induced under-catch of solid precipitation, and makes use of dynamically downscaled reanalysis to form the reference fields for data-sparse areas, and thus is considered to have a higher effective resolution. In the following, we will treat this dataset as observations and refer to it as such.

Grid-cell precipitation is an areal average of sub-grid precipitations and at a daily timescale, each value in a time series

is an accumulation over 24 hours. We upscale the fine-scale seNorge values to the coarse-scale RCM grid by calculating the weighted average over all seNorge grid cells within a given RCM grid cell, where the weights equal the proportion of each seNorge cell within the given RCM cell. The precipitation data have unit $kg \, m^{-2}$ which is approximately equivalent to mm; we then set all values less than 0.1 to 0 before other processing.

For study area, we consider the Trøndelag area in central Norway, see Figure 1. The area comprises 695 RCM grid cells

and 109 514 seNorge grid cells. The extraction of the climate change signal is performed at the RCM scale while the fine-scale daily precipitation fields are generated at nine catchments within the domain, see Figure 1 and Table 1. Two of the catchments, Krinsvatn and Oeyungen, have maritime climate while the others have continental climate. For each catchment, the modeling is performed over all seNorge grid cells within the RCM grid cells that cover the catchment, the spatial dimensions of which vary between approximately 940 and 5500 grid cells at 1 km resolution. Both historical RCM simulations and seNorge

observations are available over the time period 1957-2005. We use the time period 1957-1986 as a training period to estimate model parameters and perform an out-of-sample evaluation over the remaining 19 years 1987-2005. As a result, the training period consists of 10 950 days while the test period comprises 6 935 days.



Additionally, we use explanatory variables, or covariates, to describe the spatial variations in the statistical characteristics of the daily precipitation distributions. We consider latitude, longitude and elevation as potential geographic covariates. Elevation
information for the seNorge data is obtained from a digital elevation model based on a 100 m resolution terrain model from the Norwegian Mapping Authority (Mohr, 2009). We upscale these data in the same manner as the daily precipitation to obtain the elevation at the RCM scale. Note that this is not equal to the orography information provided by EURO-CORDEX.

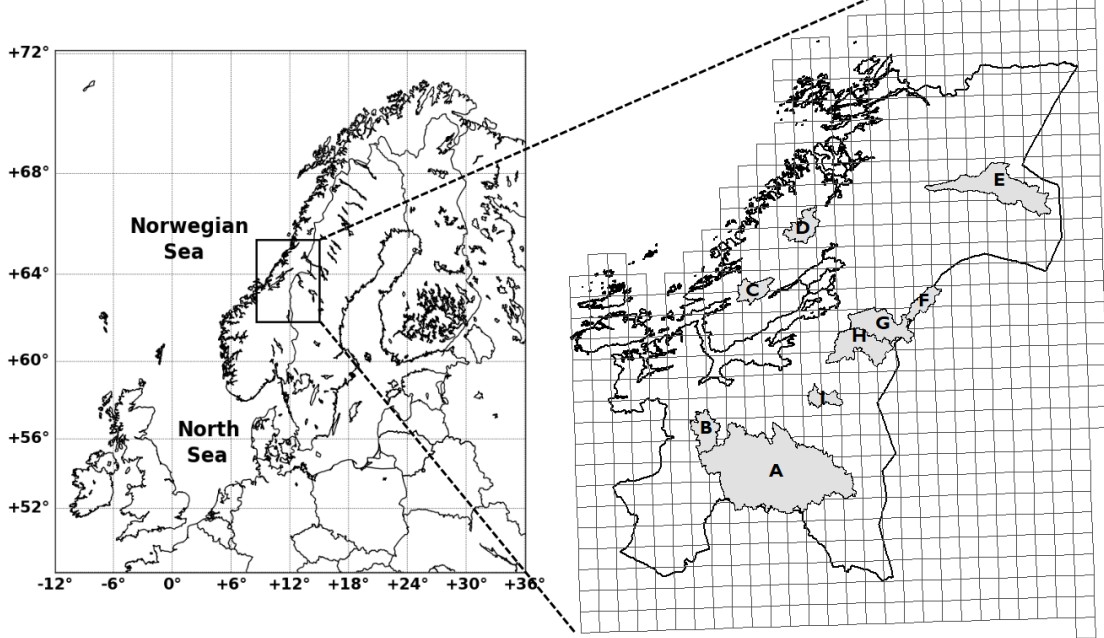

**Figure 1.** The study area is located in Trøndelag in central Norway, covering the entire Trøndelag and a small part of neighboring Sweden, and consists of 695 RCM grid cells (rectangular-like polygons) and 109 514 seNorge grid cells (within the polygons, not shown). For stochastic simulations of gridded daily precipitation, nine catchments within Trøndelag with catchment areas from 144 km$^2$ to 3084 km$^2$ (shaded in gray) are used, see also Table 1.

## 3    Methods

As mentioned in the introduction, the aim of this study is to provide realistic projections of daily precipitation at a fine spatial
scale over large areas. We apply a parametric WG approach that belongs to the class of models proposed by Wilks (1998) and Chandler and Wheater (2002). For computational feasibility, we apply the approach proposed by Serinaldi and Kilsby (2014) where a discrete-continuous distribution with a single latent field is used to simultaneously model the marginal precipitation occurrence, intensity on wet days and the space-time dependence. Specifically, we employ a combination of a latent non-stationary Gaussian space-time random field and a gamma distribution with parameters that vary in space and time, with each



**Table 1.** Characteristics of the nine catchments in Trøndelag, Norway considered in the stochastic simulations of gridded daily precipitation.

| Catchment | ID | Size (km$^2$) | Downscaling area (km$^2$) | Median elevation (m.a.s.l) |
|---|---|---|---|---|
| Gaulfoss | A | 3084 | 5479 | 734 |
| Aamot | B | 286 | 1112 | 460 |
| Krinsvatn | C | 206 | 1108 | 349 |
| Oeyungen | D | 245 | 952 | 295 |
| Trangen | E | 852 | 2327 | 558 |
| Veravatn | F | 176 | 1101 | 514 |
| Dillfoss | G | 484 | 1863 | 506 |
| Hoeggaas | H | 491 | 1853 | 505 |
| Kjeldstad | I | 144 | 940 | 578 |

model component estimated independently. The precipitation process at the RCM scale is described using a similar statistical model, and the climate change signal is added to the fine-scale model by relating the models at the two spatial scales.

### 3.1 Marginal models for precipitation occurrence and intensity

Denote precipitation occurrence in grid cell $s \in \{1,2,\ldots,S\}$ at time $t \in \{1,2,\ldots,T\}$ by $O_{st} = 1$ if there is precipitation and $O_{st} = 0$ otherwise, where $S$ denotes the number of grid cells and $T$ the number of days in a given dataset. We follow Kleiber

et al. (2012) and relate the pattern of wet and dry days to a latent Gaussian variable $W_{st}$ with mean $\mu_{st}$ and variance 1. Precipitation intensity $Y_{st}$ (i.e., the amount conditional on $O_{st} = 1$) is assumed to be gamma distributed with a constant shape $k$ and scale $\theta_{st}$ that varies over space and time, following e.g. Chandler and Wheater (2002) and Yang et al. (2005). Formally, we write

$$W_{st} = \mu_{st} + \epsilon_{st}, \quad \epsilon_{st} \sim N(0,1), \tag{1}$$

$$O_{st} = \mathbb{1}\{W_{st} > 0\}, \tag{2}$$

$$Y_{st}|O_{st} = 1 \sim \Gamma(k, \theta_{st}). \tag{3}$$

Precipitation processes often show different features depending on the time of the year, and neighboring sites tend to share similar precipitation climate. Such systematic variations are modeled by letting the parameters $\mu_{st}$ and $\theta_{st}$ of the above distributions change smoothly across time and space. We describe this through three additive components: a spatial effect, a seasonal effect and a linear climate change effect. In particular, we set


$$\mu_{st} = f_1^o(\boldsymbol{c}_s) + f_2^o(t) + f_3^o(t), \tag{4}$$

$$\log(k\theta_{st}) = f_1^g(\boldsymbol{c}_s) + f_2^g(t) + f_3^g(t), \tag{5}$$





where, in their simplest form, the three effect functions are given by

$$f_1^\zeta(\boldsymbol{c}_s) = \beta_{11}^\zeta + \beta_{12}^\zeta \mathrm{lat}_s + \beta_{13}^\zeta \mathrm{lon}_s + \beta_{14}^\zeta \mathrm{elev}_s, \tag{6}$$

$$f_2^\zeta(t) = \beta_{21}^\zeta \cos\left(\frac{2\pi\, d(t)}{365}\right) + \beta_{22}^\zeta \sin\left(\frac{2\pi\, d(t)}{365}\right) + \beta_{23}^\zeta \cos\left(\frac{4\pi\, d(t)}{365}\right) + \beta_{24}^\zeta \sin\left(\frac{4\pi\, d(t)}{365}\right), \tag{7}$$

$$f_3^\zeta(t) = \beta_3^\zeta\, y(t), \tag{8}$$

for $\zeta \in \{o, g\}$. Here, $f_1$ models the spatially varying baseline of the parameters with $\boldsymbol{c}_s$ being latitude, longitude and mean elevation of grid cell $s$. Seasonal changes are described by $f_2$, with $d(t)$ returning the calendar day of time point $t$ and $f_3$ captures potential linear trend, with $y(t)$ returning the calendar year normalized so that $\beta_3$ describes a decadal trend in the data.

This modeling framework corresponds to a GLM framework.

While the linear spatial effect function in (6) can capture the spatial variations in the occurrence at both spatial scales as well as the intensity at the RCM scale, we find that this form is too simple to capture the spatial variations in the intensity across a catchment at the finer $1 \times 1$ km scale. At the finer scale, we thus expand (6) so that the covariate contribution varies smoothly in space (Wood, 2003), expanding the model to a generalized additive model (GAM). That is, we set for the two

largest catchments A (Gaulfoss) and E (Trangen)

$$f_1^g(\boldsymbol{c}_s) = \beta_{11}^g + s_1^g(\mathrm{lat}_s, \mathrm{lon}_s) + s_2^g(\mathrm{elev}_s),$$

where $s_1$ and $s_2$ are smooth functions, and the slightly simpler

$$f_1^g(\boldsymbol{c}_s) = \beta_{11}^g + s_1^g(\mathrm{lat}_s, \mathrm{lon}_s) + \beta_{14}^g \mathrm{elev}_s$$

for the other catchments. This substantially improved the in-sample fit for all catchments. Alternatively, Kleiber et al. (2012)

propose spatially varying regression parameters.

To estimate the parameter $\mu_{st}$ of the latent Gaussian model specified in (4) and (6)-(8), we transform the data to a binary dataset with $o_{st} = 1$ if the observed value fulfils $y_{st} > 0$ and $o_{st} = 0$ if $y_{st} = 0$. We then estimate $\mu_{st}$ using probit regression with $\mathbb{P}(o_{st} = 1) = \Phi(\mu_{st})$ and $\mathbb{P}(o_{st} = 0) = 1 - \Phi(\mu_{st})$, where $\Phi$ denotes the cumulative distribution function (CDF) of the standard normal distribution. The estimation is performed using the function `glm()` in the statistical software R (R Core

Team, 2019), separately for each catchment and spatial scale. The parameters of the gamma model are estimated using only the positive values in the data set, that is, only data where $y_{st} > 0$. At the RCM scale, the gamma model is a GLM and can be estimated using `glm()`. At the seNorge scale, we employ the function `bam()` from the R package `mgcv` version 1.8-31 (Wood, 2017) so that the smooth functions $s_1$ and $s_2$ are given by thin plate regression splines as described in Wood (2003). The complexity of the spatial baseline term $f_1$ is determined by empirically assessing the spatial structure of the average in-

sample residuals over the spatial domain. Note that for a linear modeling design as in (6)-(8), `glm()` and `bam()` will return





identical estimates, insuring consistency in our estimation across the different datasets. Specifically, the inference methods return estimates of $\log(k\theta_{st})$ and $k$, from which estimates of $\theta_{st}$ can easily be derived.

## 3.2 Space-time correlation structure

The marginal models for precipitation occurrence and intensity defined in the previous section describe changes in the marginal
distributional properties across space and time. For realistic simulations of daily precipitation fields, we additionally need to account for space-time correlations of individual realizations. Here, for computational feasibility given the dimensionality of our data, we follow the approach proposed by Serinaldi and Kilsby (2014) and define a single latent Gaussian process that drives the correlation in both occurrence and intensity. We further assume that spatial and temporal correlations can be estimated separately, with the parameters of each component allowed to vary over the year to account for potential seasonality
in the correlation structure. In practice, this is performed by obtaining independent estimates for each calendar month and, subsequently, fitting a smooth function of the type given in (7) to the monthly estimates to obtain daily smoothly varying estimates. Furthermore, the correlation models are estimated independently for each catchment to account for differences between the different climatic zones.

The estimation of the correlation structure within frameworks with underlying assumptions of normality is complicated by
the shape of the precipitation distribution with its point mass in zero and the skewness of the positive part. To account for this, Serinaldi and Kilsby (2014) propose to estimate the Kendall rank correlation coefficient $\tau$ from the data (Kendall, 1945) and, subsequently, transform $\tau$ into the Pearson correlation $\rho$ by the identity $\rho = \sin(\tau\pi/2)$. For the spatial correlation structure, we use this approach to estimate the correlation between all pairs of grid cells within a catchment using the R function `cor()`. In the estimation procedure, ties are removed from the data which implies that the estimation is only based on data pairs
with two non-zero values or one zero and one non-zero value, see also the discussion in Serinaldi (2007). The R function `fit.variogram()` from the R package `gstat` (Pebesma, 2004; Gräler et al., 2016) is then employed to fit theoretical correlation functions to the empirical correlations via fitting the corresponding variogram functions. An empirical comparison of fits based on the exponential, the spherical, the Gaussian and the Matérn correlation models shows that the three-parameter Matérn model fits best in all months for all catchments.
The Matérn correlation between two grid cells with Euclidean distance $\|\boldsymbol{h}\|$ at time point $t$ is given by (e.g., Cressie and Wikle, 2015)

$$\boldsymbol{C}(\|\boldsymbol{h}\|, t) = \sigma_{0t}^2 \mathbb{1}\{\|h\| = 0\} + \sigma_{1t}^2 \{2^{\nu-1}\Gamma(\nu)\}^{-1} \{\|\boldsymbol{h}\|/\alpha_t\}^{\nu} K_{\nu}(\|\boldsymbol{h}\|/\alpha_t), \tag{9}$$

where $\Gamma$ is the gamma function and $K_{\nu}$ is the modified Bessel function of the second kind. The nugget $\sigma_{0t}^2$, partial sill $\sigma_{1t}^2$ and range $\alpha_t$ are assumed to vary over the year while $\nu$ is assumed constant. An optimal value of $\nu$ is chosen such that the sum of
squared errors of the fitted models over all 12 months is minimized. Then, a Matérn correlation function with a fixed value of $\nu$ is fitted again for each month to obtain monthly estimates of $\sigma_{1t}$ and $\alpha_t$. Here, we assume $\sigma_{0t}^2 + \sigma_{1t}^2 = 1$ to ensure that the resulting matrix is a correlation matrix.



In the literature, spatial dependencies in intensity and occurrence are commonly modeled separately assuming two latent Gaussian fields, one driving the occurrence and the other the intensity. For correlations in intensity, parametric models include the exponential (Kleiber et al., 2012) and the power exponential (Wilks, 1998; Serinaldi and Kilsby, 2014) models, as well as the simple strategy of having constant intersite correlation (Yang et al., 2005). Correlations in occurrence are more challenging to model, as appropriate transformation from binary occurrence to marginal normality is less straightforward. Wilks (1998) illustrates an empirical approach to find a link between the unobservable correlation (from a Gaussian model) and observable but unknown correlation (from a bivariate binary model) for each pair of sites. Kleiber et al. (2012) use an exponential covariance function in a similar approach. Yang et al. (2005) propose to model the number of wet sites by a beta-binomial model, and then utilize empirical conditional probabilities to allocate the positions of wet sites.

Following Serinaldi and Kilsby (2014), we introduce the short-term autocorrelation through temporal dependence in the underlying spatial random field. Here, temporal correlation is assumed to follow an autoregressive (AR) process of order one. At each grid cell, Kendall's $\tau$ is calculated for each month; the monthly value for the entire catchment is then taken as the median value over all grid cells in the catchment. Subsequently, a smooth function of the form in (7) is fitted to the 12 monthly values to obtain smoothly changing daily estimates $\hat{\rho}_t = \sin(\hat{\tau}_t \pi / 2)$. Stochastic simulation models for precipitation commonly assume autocorrelation of order one (e.g. Evin et al., 2018; Kleiber et al., 2012). However, it varies somewhat how the autocorrelation is introduced into the model. For example, Kleiber et al. (2012) include the occurrence on the previous day as a covariate in the regression models for the mean of the latent field and the parameters of the gamma intensity model.

To summarize, denote by $\boldsymbol{\epsilon}_t = (\epsilon_{1t}, \dots, \epsilon_{St})$ the vector of random noise defined in (1) in all the $S$ grid cells at time $t$. The random noise is assumed to follow a space-time correlation structure of the form

$$\boldsymbol{\eta}_t \sim N(\mathbf{0}, \boldsymbol{\Sigma}_t), \tag{10}$$

$$\boldsymbol{\epsilon}_{t+1} = \rho_t \boldsymbol{\epsilon}_t + \sqrt{1 - \rho_t^2}\, \boldsymbol{\eta}_t, \tag{11}$$

where $\boldsymbol{\Sigma}_t$ is a Matérn correlation matrix and the correlation coefficient $\rho_t$ is obtained as described above.

## 3.3 Relating models from two spatial scales

The model outlined in Sections 3.1 and 3.2 above is fitted to data at both the coarser RCM scale and the finer seNorge scale in the training period, where the significance of coefficients is tested at the 0.05 level. When fitting the marginal models to data at the coarser scale in the test period, we incorporate the training-period estimates of the coefficients into the three model components in the following manner: 1) the baseline $f_1$ is fixed to be the sum of its estimated value and the increment due to the estimated linear trend in the training period; 2) for the seasonality $f_2$ and the potential linear trend $f_3$, we use the training-period coefficients as reference, and effectively estimate the value and test the significance of the changes in these terms. In R, this could be done by using several `offset()` terms in the model formula applied in the `glm()` function. We opt for such practice in the situation where the test period directly follows the training period.





In order to obtain model parameter estimates at the finer scale in the test period, we need to relate the models at the two
scales so that model changes between the training and the test period at the coarser scale can be used to infer model changes
at the finer scale. Specifically, we may update the mean of the latent field $\mu_{st}$ in (1), the parameters of the gamma distribution
$k$ and $\theta_{st}$ in (3) and the autocorrelation coefficient $\rho_t$ in (11), while the structure of the spatial correlation matrix $\boldsymbol{\Sigma}_t$ in (10) is
assumed constant as the coarse scale data does not convey information on the fine scale spatial structures. We thus also exclude
the spatial correlation in the estimation at the coarser scale.

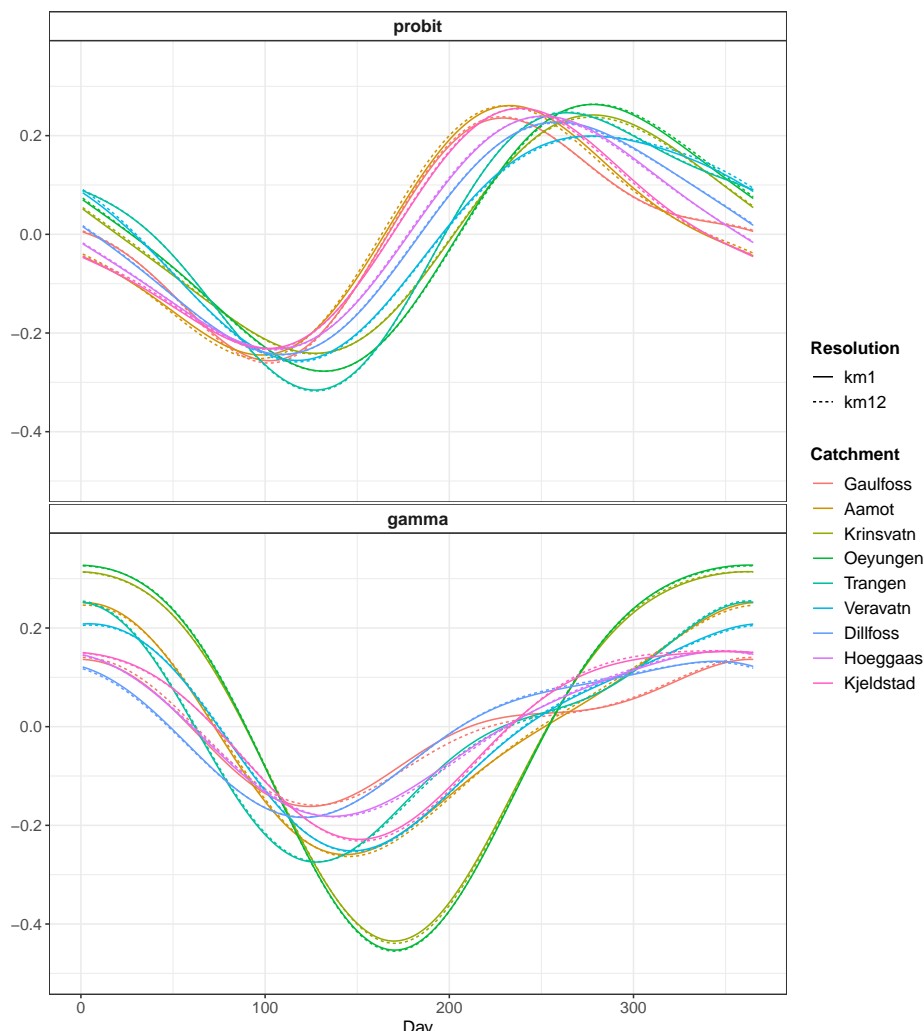

**Figure 2.** seNorge estimates of the seasonality component in (7) in the training period 1957-1986 for all catchments at both spatial scales.
Top: The estimated seasonality in the mean of the latent Gaussian field $\mu_{st}$ estimated by probit regression. Bottom: The estimated seasonality
in the mean of the gamma distribution $\log(k\theta_{st})$ estimated within a GLM/GAM framework.





For $\mu_{st}$ and $\log(\theta_{st}) = \log(k\theta_{st}) - \log(k)$, we may update each of the terms in (4) and (5), respectively. Here, the seasonality (7) and the potential linear trend component (8) of $\mu_{st}$ (and similar for $\log(\theta_{st})$) are adjusted so that the average adjustment over all the time points in the test period $\bar{\mu}_{s.}^{a}$ fulfils

$$\bar{\mu}_{s.}^{a} = \bar{\mu}_{s.}^{tr} + (\bar{\mu}_{r.}^{te} - \bar{\mu}_{r.}^{tr}),$$

where $te$ indicates the test period, $tr$ indicates the training period, and $s$ is a fine scale grid cell located within a coarse scale
grid cell $r$.

Figure 2 shows the training-period estimates of the seasonality component given in (7). While the seasonality patterns vary substantially across the different catchments as well as between the two model parts, the estimates are very consistent across the two spatial scales. We thus infer seasonality components for the fine scale during the test period by updating the fine scale components from the training period according to the estimated changes between the training and the test period at the coarse
scale. We see the same patterns for the trend coefficient in (8), see Table 2. The trend coefficient and the correlation coefficient $\rho_t$ are thus updated in the same manner as the seasonality component. Finally, the shape parameter of the gamma distribution $k$ may be updated so that the ratio of the estimates in the training and the test period at the fine scale equals the ratio of the two estimates at the coarser scale.

**Table 2.** The estimated trend coefficient in (8) for each catchment based on data from 1957-1986 for $\mu_{st}$ in the probit model (left) and $\log(\theta_{st})$ in the gamma model (right). Estimates are given for both 1 km seNorge data and seNorge data upscaled to 12 km resolution.

| | $\mu_{st}$ | | $\log(\theta_{st})$ | |
| | seNorge | seNorge | seNorge | seNorge |
| Catchment | $1 \times 1$ km | $12 \times 12$ km | $1 \times 1$ km | $12 \times 12$ km |
|---|---|---|---|---|
| Gaulfoss | 0.002 | 0.002 | -0.003 | -0.004 |
| Aamot | 0.009 | 0.011 | 0.046 | 0.045 |
| Krinsvatn | 0.035 | 0.036 | 0.023 | 0.020 |
| Oeyungen | 0.020 | 0.019 | 0.045 | 0.047 |
| Trangen | 0.001 | 0.000 | 0.038 | 0.039 |
| Veravatn | 0.051 | 0.049 | 0.016 | 0.018 |
| Dillfoss | 0.022 | 0.020 | -0.026 | -0.025 |
| Hoeggaas | 0.010 | 0.010 | -0.024 | -0.023 |
| Kjeldstad | -0.003 | -0.003 | 0.013 | 0.013 |

In Section 4 various versions of the method are compared, where individual model components are either updated according
to information based on an RCM output, or assumed stationary over the entire time period.

### 3.4    Daily fine-scale precipitation generator

With the adjustments described above, the marginal models and the space-time Gaussian random field together form a precipitation generator for use on the fine-scale grid in the test period. The parameters of the generator are obtained using seNorge





data in the training period and adjusted based on RCM data spanning both training and test periods. Assume we want to simu-
late data at all grid cell locations $s \in \{1, \dots, S\}$ and time points $t \in \{1, \dots, T\}$, a total of $S$ locations and $T$ time points. Data
simulation from the generator consists of the following steps, with the superscript $a$ indicating adjusted parameter estimates:

1. For each time point $t$, spatially correlated but temporally independent random vectors $\boldsymbol{\eta}_t^*$ of size $S$ are drawn from the
   multivariate Gaussian distribution with mean vector $\mathbf{0}$ and correlation matrix $\hat{\boldsymbol{\Sigma}}_t$ specified by the Matérn correlation
   function, i.e. $\boldsymbol{\eta}_t^* \sim N(\mathbf{0}, \hat{\boldsymbol{\Sigma}}_t)$.

2. Temporal correlation is introduced by setting $\boldsymbol{\epsilon}_{t+1}^* = \hat{\rho}_t^a \boldsymbol{\epsilon}_t^* + \sqrt{1 - (\hat{\rho}_t^a)^2}\, \boldsymbol{\eta}_t^*$.

3. At grid cell $s$ and time $t$, the probability of precipitation is $\hat{p}_{st}^a = \Phi(\hat{\mu}_{st}^a)$. The precipitation amount is set as $y_{st}^* = 0$ if
   $\Phi(\epsilon_{st}^*) \leq 1 - \hat{p}_{st}^a$ and $y_{st}^* = \Gamma^{-1}((\Phi(\epsilon_{st}^*) - (1 - \hat{p}_{st}^a))/\hat{p}_{st}^a; \hat{k}^a, \hat{\theta}_{st}^a)$ otherwise.

That is, as mentioned above, the fine-scale spatial correlation structure described by $\hat{\boldsymbol{\Sigma}}_t$ is the single part of the model that is
not adjusted based on information from the RCM.

## 3.5 Reference method

To assess the performance of our method, we use the empirical quantile delta mapping method as a reference. The RCM
outputs of approximately $12 \times 12$ km resolution are first re-gridded to the $1 \times 1$ km seNorge grid using bilinear interpolation,
as implemented in the R package `akima` version 0.6-2 (Akima and Gebhardt, 2016). Wet-day correction is applied prior to
bias correction of precipitation amount, as RCM outputs tend to give more rainy days than the observed (Frei et al., 2003).
Specifically, a threshold value is determined such that the wet-day frequency in the re-gridded RCM dataset is equal to that
in the seNorge dataset for the training period; precipitation values below the threshold value are set to zero for both training
and test periods. Correction of precipitation amount in the test period is carried out using the empirical quantile delta mapping
method proposed by Cannon et al. (2015), where the relative changes in the precipitation quantiles projected by an RCM from
the training period to the test period are explicitly preserved. The method belongs to the class of widely used empirical quantile
mapping methods (EQM), and is applied here at individual seNorge grid cells on monthly basis.

## 3.6 Evaluation methods

An evaluation and comparison of the different approaches is performed by comparing various aspects of the resulting datasets.
For an overall ranking of the approaches, we employ the proper evaluation metric integrated quadratic distance (IQD) that
compares the full distributions of observed and modeled precipitation (Thorarinsdottir et al., 2013). That is, denote by $F$ the
empirical cumulative distribution function (ECDF) of seNorge precipitation over all time points in the test set at a given grid
cell and by $G$ the corresponding ECDF from one of the modeling approaches. The distance between $F$ and $G$ as measured by





the IQD is then given by

$$d(F,G) = \int\limits_{-\infty}^{+\infty} (F(x) - G(x))^2 \mathrm{d}x.$$

The overall performance of the model at a catchment is then calculated as the average IQD over all grid cells in the catchment
area with a lower value indicating a better performance. The IQD fulfills the property that the true data generating process is expected to obtain an IQD value of 0 when compared against ECDFs based on data samples of any size. It is thus an appropriate metric for ranking competing methods (Gneiting and Raftery, 2007; Thorarinsdottir et al., 2013). For the WG approach, we can easily obtain a precise approximation of the marginal distribution in each grid cell by simulating multiple realizations from each daily distribution. For the EQM approach, however, the marginal distribution in a grid cell is estimated by combining one
value for each day in the time period of interest.

For an improved understanding of the behavior of the models, we further perform several empirical diagnostics. To analyze the marginal distributions at each grid cell, we compare means of daily precipitation, wet-day frequency given by the number of wet days, wet-day intensity as measured by the mean and standard deviation of the precipitation on wet days only, and representation of heavy precipitation as measured by the 95th percentile of positive precipitation. Diagnostics of the temporal data
structure are performed by assessing dry-wet temporal patterns and seasonal patterns of temporal autocorrelation coefficients, while empirical functions of the Pearson's correlation as a function of distance are used to perform spatial data diagnostics.

## 4   Results

We perform model inference using data from 1957-1986 and infer climate change information by comparing the coarse scale RCM data from the two time periods 1957-1986 and 1987-2005. Simulations of fine scale precipitation for the test set 1987-
2005 are then compared against the seNorge data for the test period 1987-2005.

We consider three versions of the WG method, where we include varying degrees of climate change information derived from the RCM data. A stationary version, denoted by WGs, assumes that trends estimated for the seNorge data in the training period continue into the test period with the remaining model components fixed at their estimates in the training period. That is, no RCM information is used. A version denoted by WG1.1 and WG2.1 for RCM information derived from RCM1 and RCM2,
respectively, includes climate change information from the RCM in the seasonality and trend components of the gamma model for precipitation amount on wet days. Finally, a version denoted by WG1.2 and WG2.2 for RCM information derived from RCM1 and RCM2, respectively, includes climate change information from the RCM in the seasonality and trend components of both the gamma model and the probit model for precipitation occurrence. The various WG methods are compared against the reference method in Section 3.5 denoted EQM1 and EQM2 derived from RCM1 and RCM2, respectively, as well as a
simple method that uses the empirical distributions of the fine scale seNorge data in the training period directly as predictions for the corresponding empirical distributions of the fine scale seNorge data in the test period.





## 4.1 Marginal performance

**Table 3.** Integrated quadratic distance (IQD) values comparing simulated and seNorge distributions over all days in 1987-2005. The results are averaged over all $1\times1$ km grid cells in each catchment. The simple method seNorge uses the daily values over the period 1957-1986 as a prediction, WGs assumes trends estimated for 1957-1986 continue in 1987-2005, WG1.1 and WG2.1 include seasonality and trend estimates from RCM1 and RCM2, respectively, in the gamma model, while for WG1.2 and WG2.2, RCM information is included in both the gamma model and the probit model. Results of the reference method are denoted EQM1 for RCM1 and EQM2 for RCM2. The best method for each catchment is indicated in bold.

| Catchment | seNorge | WGs | WG1.1 | WG2.1 | WG1.2 | WG2.2 | EQM1 | EQM2 |
|---|---|---|---|---|---|---|---|---|
| Gaulfoss | 3.46 | 3.99 | 2.87 | 3.10 | 3.91 | 2.97 | 3.73 | **2.80** |
| Aamot | 2.23 | **1.64** | 2.90 | 2.37 | 2.37 | 2.86 | 2.67 | 2.33 |
| Krinsvatn | 8.18 | 1.94 | 3.02 | 1.96 | 2.54 | **1.79** | 12.27 | 7.62 |
| Oeyungen | 5.52 | 5.94 | 7.14 | 7.46 | **4.90** | 6.44 | 11.20 | 4.91 |
| Trangen | 9.37 | 5.56 | **5.12** | 5.50 | 6.12 | 5.49 | 10.72 | 7.84 |
| Veravatn | 11.26 | 2.66 | 2.37 | 2.24 | 2.77 | **2.22** | 15.45 | 8.12 |
| Dillfoss | 5.17 | 6.59 | 4.73 | 4.27 | 6.97 | 4.23 | 5.58 | **3.05** |
| Hoeggaas | 2.65 | 5.84 | 3.54 | 3.21 | 6.15 | 3.17 | 3.21 | **1.46** |
| Kjeldstad | 6.96 | 6.71 | 4.32 | 4.00 | 6.51 | 3.96 | 7.38 | **3.50** |
| Overall | 4.88 | 4.50 | 3.60 | 3.65 | 4.61 | **3.54** | 5.82 | 3.83 |

We evaluate the marginal performance of the simulations by comparing empirical distributions of simulations and observations over all time points in the test set. Specifically, we compare the empirical distribution of the seNorge data in every $1\times1$

km grid cell to simulations for that same grid cell using the IQD. The average IQD values over all grid cells in each catchment are given in Table 3. Overall, the WG methods that include RCM information perform better than the stationary approach which again outperforms using the historical data directly. The WG simulations have better performance than EQM for both RCM1 and RCM2. The best performing simulation is WG2.2, where both the gamma model for precipitation amount and the probit model for the wet frequency are updated with climate change information from RCM2. EQM based on RCM2 performs

quite well, while EQM based on RCM1 yields the worst performing results.

The IQD values in Table 3 vary substantially across the simulation methods for individual catchments. To investigate this further, we take a closer look at the trend coefficient estimates, as the estimated changes in seasonality are quite stable across catchments for a given RCM and model component (results not shown). The estimates of the trend coefficient in (8) based on the seNorge training data from 1957-1986 are given in Table 2 in Section 3.3 above. For the probit model, the trend estimates

are positive in all but one catchment, the small inland catchment Kjeldstad, where a small negative trend is estimated. As a result, the probability of precipitation is expected to increase over time. The rate of the increase varies substantially for the different catchments, ranging from 0.001 in Trangen to 0.051 in Veravatn. For the gamma distribution, the trend coefficient estimates are highly varying across catchments, with negative estimates for three catchments and positive estimates for six cathcments, indicating no consistent trend pattern in the amount of daily precipitation on wet days. When fitting these models

to the RCM data in the training period, we found insignificant trend estimates for the probit model in seven catchments based





on RCM1 and five based on RCM2, while the number of cases for the gamma model is six based on RCM1 and four based on RCM2.

The estimated changes in trend coefficients at $12 \times 12$ km scale between training and test period are listed in Table 4. The zeros in the table indicate that the changes are not significantly different from 0 at the 0.05 level. The seNorge estimates for
the probit model are mostly positive, corresponding to a higher trend estimate in the test period than the training period. The estimates based on RCM1 are consistently negative, while no change is estimated based on RCM2 except for Aamot. For the gamma model, approximately as many positive and negative values are observed, while estimates in all catchments are positive by both RCMs. Note that the stationary simulation WGs assumes the same trends in the training and test periods, corresponding to values of 0 in Table 4.

**Table 4.** Estimated changes in the trend coefficient in (8) between the training period 1957-1986 and the test period 1987-2005, for $\mu_{st}$ in the probit model (left) and $\log(\theta_{st})$ in the gamma model (right). Estimates for three different data sources at 12 km resolution are shown: upscaled seNorge data and two RCM outputs.

| | $\mu_{st}$ | | | $\log(\theta_{st})$ | | |
|---|---|---|---|---|---|---|
| Catchment | seNorge | RCM1 | RCM2 | seNorge | RCM1 | RCM2 |
| Gaulfoss | 0.026 | -0.022 | 0.000 | 0.034 | 0.040 | 0.025 |
| Aamot | 0.000 | -0.018 | 0.013 | -0.081 | 0.034 | 0.013 |
| Krinsvatn | -0.014 | -0.044 | 0.000 | -0.043 | 0.037 | 0.014 |
| Oeyungen | 0.019 | -0.044 | 0.000 | -0.103 | 0.021 | 0.021 |
| Trangen | 0.080 | -0.012 | 0.000 | -0.012 | 0.020 | 0.000 |
| Veravatn | -0.093 | -0.029 | 0.000 | 0.039 | 0.018 | 0.023 |
| Dillfoss | -0.021 | -0.033 | 0.000 | 0.069 | 0.031 | 0.028 |
| Hoeggaas | 0.000 | -0.033 | 0.000 | 0.057 | 0.039 | 0.034 |
| Kjeldstad | 0.039 | -0.022 | 0.000 | 0.038 | 0.040 | 0.040 |

The simulations WGs, WG1.1 and WG2.1 share the same probit model for precipitation occurrence, while the gamma model for the precipitation amount differ. For the gamma model, five catchments have a strong positive climate change signal according to the upscaled seNorge data, where both RCMs project a change in the same direction. Looking at the IQD values in Table 3 we see this translates directly into lower IQD values compared to the WGs simulations. IQD values are higher than WGs in the three catchments closest to the coast (Aamot, Krinsvatn and Oeyungen), where both RCMs project a positive
change against the observed negative change. For Trangen, WG2.1 and WGs have similar IQD values because they both apply no change in the trend. In general, both RCMs provide useful climate change information for the gamma model which makes the overall performance of WG1.1 and WG2.1 better than WGs.

Similar effect can be seen when comparing the IQD values for Gaulfoss, Trangen and Kjeldstad based on the simulations WG1.1 and WG1.2. While these two simulations share the same gamma model, WG1.1 assumes a stationary probit model and
WG1.2 applies climate change information from RCM1 on the precipitation occurrence. Here, the climate change estimates from RCM1 are negative, going in the opposite direction as for the upscaled seNorge data, and accordingly WG1.2 is worse than WG1.1 which assumes no change in the trend. The negative change applied in WG1.2 in Hoeggaas can also relate to





the reduced performance compared with WG1.1. In Veravatn and Dillfoss, however, the estimates based on RCM1 are in the
same direction as the observed but this somehow does not translate into a better performance of WG1.2. For Aamot, where
no change is estimated by the upscaled seNorge data, a negative change by RCM1 seems to make WG1.2 better than WG1.1,
and a positive change by RCM2 makes it the only catchment where WG2.2 is worse than WG2.1. In the other catchments,
WG2.2 is slightly better than WG2.1 given that they both apply no change in the trend of the probit model; this indicates that
the changes in the seasonality projected by RCM2 is generally reasonable only the effect seems limited in most catchments.

Further analysis of the marginal performance of four of the simulations as well as the seNorge reference is shown in Figure 3
for the largest catchment Gaulfoss. The leftmost plot in Figure 3(a) shows that the frequency of wet days for the seNorge data
is generally lower in the training period than the test period. This again results in a significant bias in the overall mean, see
Figure 3(b), while the general correspondence between the amount distributions on wet days is quite good. Here, the IQD
value is 3.46 for seNorge, 3.99 for WGs, 3.10 for WG2.1, 2.97 for WG2.2 and 2.8 for EQM2. WG2.1 and WG2.2 share the
same distribution for the precipitation amount on wet days, and given that RCM2 projects zero change in the trend of the
probit model, performance of the two simulations is different solely due to the different seasonality which again is minimal,
see Figure 3(a). While EQM2 has the lowest IQD value, it appears that this method overestimates the wet frequency, see
Figure 3(a), the spread on wet days Figure 3(d) and thus also the 95th percentile on wet days Figure 3(e). However, the IQD
score is less sensitive to these errors than to erroneous overall mean.

### 4.2   Spatial and temporal correlation structure

The spatial correlation structure at $1 \times 1$ km scale cannot be inferred from the $12 \times 12$ km RCM data and we thus assume that the
fine-scale spatial correlation estimated based on the training data also holds for the test data. This is assessed in Figure 4 for the
largest catchment Gaulfoss and in Figure 5 for the smallest catchment Kjeldstad. The Matérn correlation function estimated
based on the training data appears to capture the overall structure of the test data, indicating no large deviations in spatial
structure between the two time periods. However, there are some smaller deviations, indicating smaller changes in the seasonal
pattern of the spatial structure. In particular, the estimated correlation is slightly higher than the observed in February and
somewhat lower in fall, especially at Kjeldstad. For both catchments, the largest spread of the daily estimates of the correlation
function is in the spring months of April and May.

The spatial structure of the EQM simulation differs somewhat from that of the data. The correlation is too strong in the
winter months of December, January and February, and too weak in June. It further appears that the EQM is more successful
in modeling the spatial correlation of the data from the larger catchment Gaulfoss than the data from the small catchment
Kjeldstad, which area of 144 km$^2$ is approximately 5% of the area of Gaulfoss at 3084 km$^2$.

In order to assess the temporal correlation structure of the various simulations, first consider the two-day dry/wet patterns
shown in Figure 6. For the inland catchment Gaulfoss, the proportions of two consecutive dry days and two consecutive wet
days is approximately equal in the test set, while the training set has fewer instances of two consecutive dry days with a
corresponding increase in two consecutive wet days. The proportions of two consecutive dry or wet days for the simulations
are mostly in between the values for the seNorge training and test set, except for EQM2 which has the highest frequency of wet







(a) Relative bias in frequency of wet days

(b) Relative bias in overall mean

(c) Relative bias in mean on wet days

(d) Relative bias in SD on wet days

(e) Relative bias in 95th percentile on wet days

**Figure 3.** Relative bias in various marginal summary statistics at 1×1 km scale in the largest catchment Gaulfoss. The observed seNorge data in the training period 1957-1986, the stationary WGs simulation and three simulations using climate change information from RCM2 are compared against the seNorge data in the test period 1987-2005.


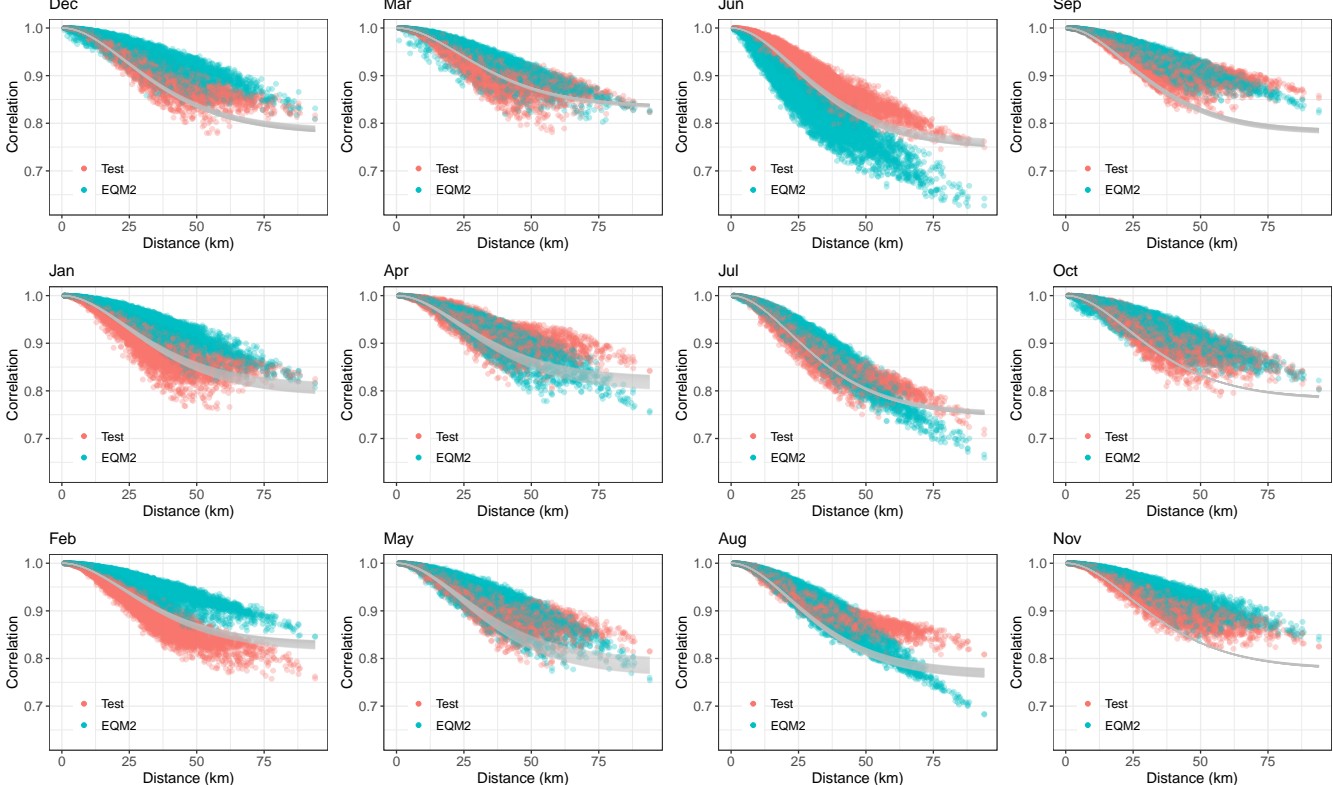

**Figure 4.** Empirical spatial correlation of precipitation amount at the catchment Gaulfoss for each month of the year. Results are shown for the seNorge data in the test period 1987-2005 (red dots) and for the EQM simulation based on RCM2 (cyan dots). The Matérn spatial correlation estimated with the WG method based on seNorge data in the training period 1957-1986 is indicated in gray with the width of the bar indicating the spread of the daily estimates within the month.

days, see also Figure 3(a). At the coastal catchment Oeyungen nearly 50% of all the two-day patterns observed in the training period, and over 50% in the test set, are two consecutive wet days. Here, all the simulations yield a lower proportion of two consecutive wet days than the observed test data, while the proportions of pairs with one wet day and one dry day is higher.

The results shown here for the WG method are based on a single simulation for each model version. We found that these results may very slightly between realizations from the same model (results not shown).

The temporal correlation applied in the daily fine-scale precipitation generator for the test period is assessed in Figure 7. As described in Section 3.2, the short-term autocorrelation of the WG method is introduced through the temporal dependence in the underlying spatial random field. Data at both spatial scales have the same temporal dimensionality, we thus assume that the

fine-scale temporal correlation coefficients $\rho_t$ can be updated by the changes projected by an RCM between training and test periods. Estimates based on seNorge data in the training period indicate higher temporal dependence in spring and winter and lower in summer. In the test period, dependence becomes lower in spring and summer and higher in October and November.





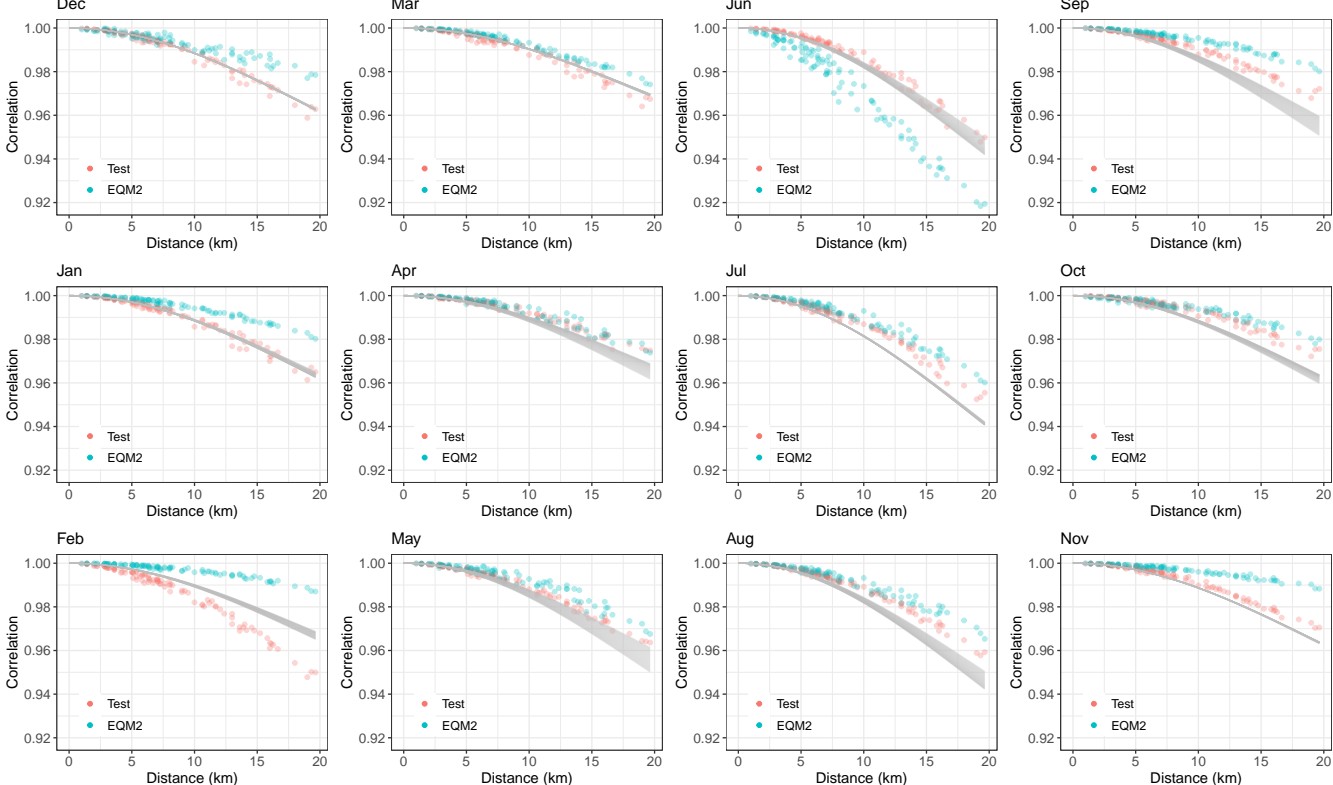

**Figure 5.** Empirical spatial correlation of precipitation amount at the catchment Kjeldstad for each month of the year. Results are shown for the seNorge data in the test period 1987-2005 (red dots) and for the EQM simulation based on RCM2 (cyan dots). The Matérn spatial correlation estimated with the WG method based on seNorge data in the training period 1957-1986 is indicated in gray with the width of the bar indicating the spread of the daily estimates within the month.

The changes in spring are generally not realistically projected by RCMs, except for RCM2 in Trangen, while the changes in summer and early winter are better captured by RCM2 than RCM1 in most catchments.

**5 Conclusions and discussion**

This paper proposes a two-step stochastic downscaling and bias-correction approach for future projection of daily precipitation. In a first step, a stochastic weather generator for a high-resolution grid is developed using a historical gridded observation-based data product. In a second step, the weather generator is calibrated for a future climate by using only the projected changes between a historical reference period and a future period based on a coarser scale RCM. In the current application, the

observation-based data product is available on a $1 \times 1$ km grid and the climate change information stems from an RCM on a $12 \times 12$ km grid. In this setting, there appears to be good correspondence between catchment-scale seasonality and linear trend patterns at the two spatial resolutions, making the transformation of information between the two scales feasible.





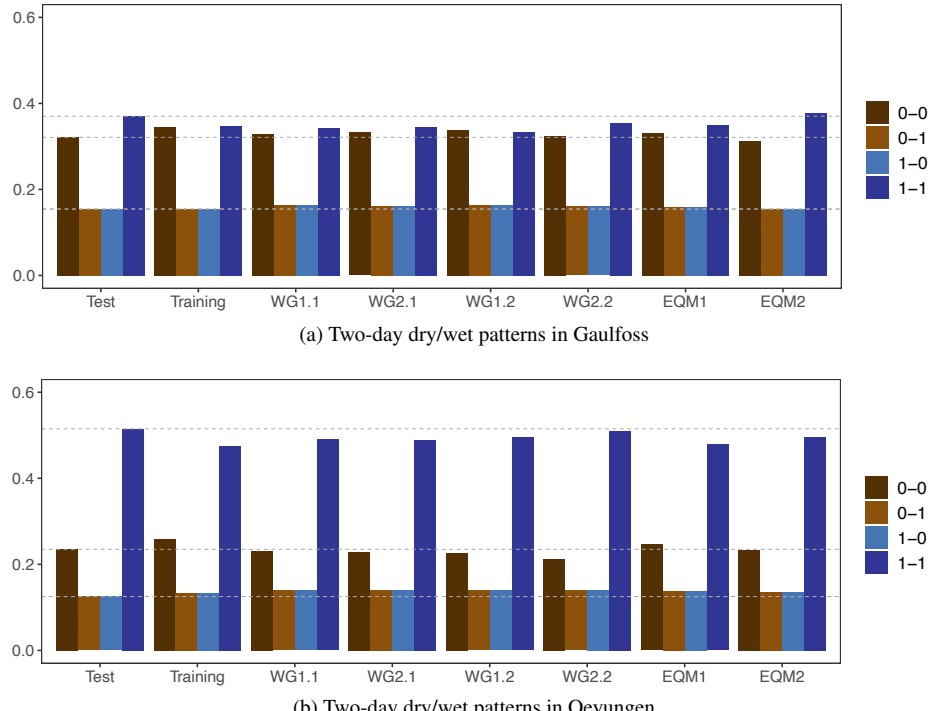

**Figure 6.** Proportion of different two-day dry/wet patterns for the seNorge data in the training period 1957-1986 and the test period 1987-2005, as well as for six different simulations of the test period. The results are aggregated over all grid cells in the catchments Gaulfoss (top) and Oeyungen (bottom). Dry days are indicated with 0 and wet days with 1. For ease of interpretation, horizontal dashed lines are drawn at the levels of the test set.

The proposed WG approach is applied to data from nine hydrological catchments in central Norway, with each study area ranging in size from approximately 1000-5500 km$^2$, and compared against the empirical quantile delta mapping method that belongs to the EQM approach and a simple persistence reference method. The methods are trained on daily data from 1957-1986 and tested on out-of-sample data from 1987-2005. Based on an evaluation of the resulting marginal distributions, the WG method overall outperforms the EQM approach, both in terms of the IQD score and based on empirical assessment of marginal summary statistics. However, all the simulation methods show large variations in the performance between individual catchments. The WG method furthermore yields realistic temporal and spatial correlation structure.

The historical RCM runs used here are available until 2005, and the observation-based data product is available from 1957, yielding a dataset with 49 years of data. With 30 years of data used to train the models, this leaves only 19 years of data for the out-of-sample evaluation. With only 19 years of data in the test period, we may expect to see some effects of natural variability. Looking at the linear trend coefficient in the probit model, it seems that the seNorge data upscaled to 12 km resolution are generally able to capture the change that there are proportionally more wet days in the test period than in the training period, while the RCM data either project strong negative changes or simply no change in most catchments. For the gamma model, both RCMs seem to have projected correct changes in the trend and seasonality. For this reason, it cannot be expected that the





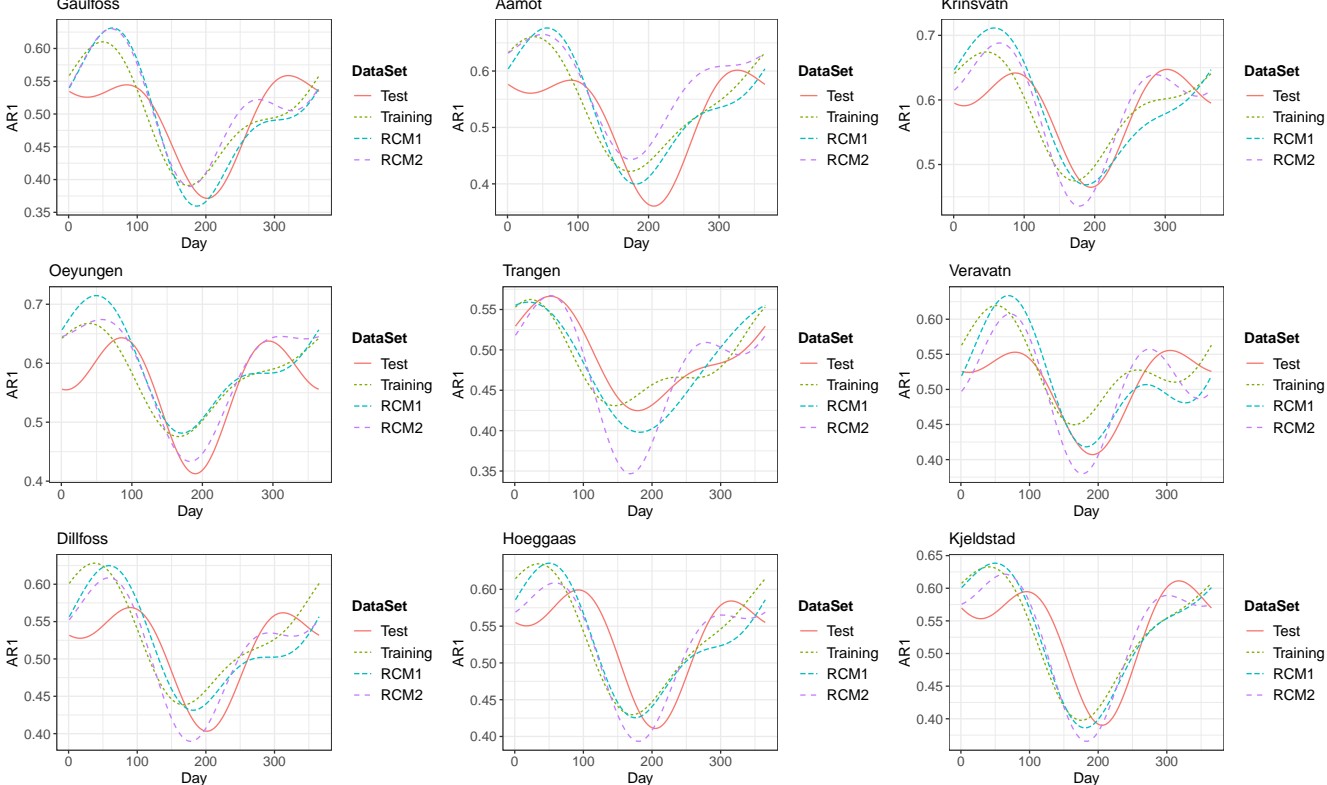

**Figure 7.** Smoothly changing daily estimates of the correlation coefficient $\rho_t$ in (11) for each catchment, estimated based on the seNorge data in the training period 1957-1986 (green dotted lines), inferred by adding the climate change information from RCM1 (cyan dashed lines) and RCM2 (purple dashed lines) for the test period 1987-2005, and as reference the values estimated based on the seNorge data in the test period 1987-2005 (red solid lines).

RCM runs perfectly reproduce the changes in precipitation patterns in the finer-scale seNorge data. Nevertheless, we see that all versions of the WG method yield better performance than the marginal persistence reference method based on seNorge data from 1957-1986, and including RCM information improves upon the stationary WG approach. Furthermore, the transparent
way in which the RCM information is included in the WG simulations allows for a direct assessment of this information and its plausibility (Maraun et al., 2017).

While the application in this paper focuses on climate projections, the modeling framework proposed here provides a more general approach for computationally efficient stochastic downscaling of precipitation. Other potential applications include seasonal and decadal weather and climate predictions. The availability of computationally efficient downscaling methods is
especially important in settings where large ensembles are needed in order to achieve prediction skill, see e.g. Smith et al. (2019).



*Code availability.* Code is available upon requests from the authors.

*Data availability.* The seNorge version 2018 data is available at http://thredds.met.no/thredds/catalog/senorge/seNorge_2018/. The two RCM datasets from the EURO-CORDEX-11 ensemble are available at https://www.euro-cordex.net/060378/index.php.en.

*Author contributions.* All the authors defined the scientific scope of this study together. TLT and QY formulated the methodology of the paper. QY prepared the R code for the statistical modeling, simulations and evaluations of the proposed method. TLT provided support in many parts of the R code. WKW provided the results of the reference method and Figure 1. TLT and QY contributed to the write-up of the manuscript. All the authors provided ideas, suggested improvements during the entire process of conducting the research.

*Competing interests.* No competing interests are present.

*Acknowledgements.* This work was supported by the Research Council of Norway through project nr. 255517 "Post-processing Climate Projection Output for Key Users in Norway". The work of Thordis Thorarinsdottir was additionally supported by the Research Council of Norway through project nr. 309562 "Climate Futures".



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
