# Peer review of "Bridging the scale gap: Obtaining high-resolution stochastic simulations of gridded daily precipitation in a future climate"

_Hydrology and Earth System Sciences, 2020_

## Author Comment (AC1)

**Bridging the scale gap: Obtaining high-resolution stochastic simulations of gridded daily precipitation in a future climate (Manuscript number HESS-2020-637)**

Qifen Yuan, Thordis L. Thorarinsdottir, Stein Beldring,
Wai Kwok Wong, Chong-Yu Xu

**Response to Referee 1**

We thank the reviewer for his/her positive and insightful comments on the manuscript. Below is our response to the issues raised in the review.
* * *
**Referee 1**

This is a very meaningful study that puts forward a combination of statistical methods and approaches to produce fine scale resolution rainfall estimates for catchments (small to medium/large-ish sizes). It combines aspects of the weather-generator type approach with spatial statistics, acknowledging the need to also consider temporal dependencies and structures.

The language and graphics are clear and concise, explicit formulas are provided, accompanied by relevant references to underpinning methods and reference to used R-packages/methods.

Whilst I think that there are a one or two of issues to address method wise before it can be applied in a climate change context, I think it is a meaningful suggestion to build on. Overall, in my opinion this manuscript offers a very meaningful contribution to the peer-literature on statistical downscaling and merits publication with only minor edits.

**Reply**: Thank you for your positive evaluation of our work. We will modify the manuscript following your suggestions. Please find details below.

**General comments**

Given the focus of this paper, I suggest a few changes that would make the paper more accessible to a wider audience that should pay attention to this work but would find it

difficult to detangle the methodology (the paper has a very strong statistical flavour and assumes that readers are familiar with statistical terminology). *Firstly*, I suggest that you put in a conceptual graphic, that outlines the different components in our analysis and how they fit together, this will help the reader understand how the different model parts fits together, and where the 'change' is applied (change between the two time periods). *Second*, I think it will help the reader if you outline when you expect the model to work well, and when you expect there to be difficulties in the methods section. Some of the performance issues discussed in the discussion section are entirely predictable and I think it will help the reader understand the method better if you have a paragraph that speaks about the expected strengths and expected weaknesses before you conduct the experiments, this will get the reader introduced to how the method works, and then the results will make more sense. I think the manuscript will be more accessible and thus more successful if it embraces a wider audience.

**Reply**: Thank you for your suggestions. Firstly, we will include a conceptual graphic in the introduction section, see Figure 1 below. Second, we will add a paragraph in the introduction section where we introduce our approach (after line 55) where we state the two main strengths of our method (that it is stochastic in nature and that climate change is added to the simulations in a fully transparent manner) as well as its two main assumptions (that the RCM correctly captures the climate change signal in the model components and that climate change effects are transferrable between the two scales).

[Figure]

Figure 1: The proposed two-stage weather generator approach for simulations of fine-scale daily precipitation in a future climate.

One aspect not discussed in this paper is the applicability in a climate change context. Whilst the two time periods do exhibit some climatological differences, changes that one might expect in 20-50 years time are likely to be more pronounced. In this context, you may need to include a representation of oscillatory behaviour in the spatial dependency structure (in addition to a seasonal representation). This may not be the case for Norway, but other regions with more variable rainfall climatologies (strongly influenced by e.g. ENSO or IDO ... or any other large-scale climate oscillation or teleconnection pattern. I think it would be very meaningful if you could elaborate on this need in your discussion section, and perhaps propose a couple of tests to see if this is necessary, e.g. compare spatial correlation in subsets of RCM rainfall fields that capture certain seasonal/oscillation modes – are these still similar enough (how to judge?) to what we see in the coarse resolution observed rainfall fields?

**Reply**: This is indeed a very important discussion point for a practical application of our method in climate change impact studies. We will add the following paragraph to the discussion section (after line 411): " In our case study, the training and the test period are two consecutive time periods. However, in climate change impact studies, there is commonly a large gap on the order of decades between the historical period and the future period of interest. In this case, it may be necessary to expand our proposed model to also account for large-scale climate oscillation or teleconnection patterns, such as the El Niño-Sourthern Oscillation (ENSO) and the Indian Ocean Diopole (IOD), particularly in regions where rainfall climatologies are dominated by such patterns (e.g. Wu et al., 2003; Andreoli and Kayano, 2005). In such cases, specific components of the model, e.g. the spatial correlation structure, may need to be estimated depending on both seasonal variation and oscillation modes. To assess this, the parameter estimation procedure can be extended to obtain separate estimates for both months and oscillation modes. The series of parameter estimates can then be assessed for seasonal and oscillation dependence using standard regression techniques."

**Four smaller issues**

(1) I think it is always useful to be critical of the RCM – hence I think the authors should point out the need to demonstrate that the RCMs selected for the analysis do indeed capture the spatial and temporal patterns of the variable of interest on its resolution before downscaling (where the downscaling is drawing on a relationship derived from observed data).

**Reply**: It is useful to check if the spatial and temporal patterns of the RCM outputs are realistic at its own scale. In a pre-analysis step, we compared three summary statistics at individual grid cells between the RCM data and the upscaled seNorge data (i.e. the approximate true climate at the RCM scale) for the training period 1957-1986. We can see in Figure 2 below, that the overall mean and the mean on wet days of the daily precipitation from the two RCMs have similar spatial patterns as the upscaled seNorge, although with an underestimation in the coastal area and colder seasons. The frequency of wet days is substantially overestimated by RCMs as expected, but the underlying spatial pattern resembles the upscaled seNorge data, see Figure 3 below.

[Figure]

(a) Overall mean

[Figure]

(b) Mean on wet days

Figure 2: Overall daily mean and the mean on wet days in the training period 1957-1986 calculated over all days (Annual) as well as days by season: Spring (MAM), Summer (JJA), Autumn (SON) and Winter (DJF), based on three datasets at 12 km resolution: the upscaled seNorge data and two RCMs.

[Figure]

Figure 3: Frequency of wet days in the training period 1957-1986 calculated over all days (Annual) as well as days by season: Spring (MAM), Summer (JJA), Autumn (SON) and Winter (DJF), based on three datasets at 12 km resolution: the upscaled seNorge data and two RCMs.

(2) Using empirical scaling (or decile scaling) as a reference is meaningful because that is a very common method. However, many water-related studies would use daily scaling rather than monthly scaling as you are draping a scaling coefficient of monthly resolution onto a variable with daily resolution, then evaluating on the daily resolution. The comparison seems a little unfair. I see no need to redo your analysis, but I would point out how the monthly scaling could impact the comparison.

**Reply**: The quantile delta mapping method can be implemented on different time scales (Cannon et al., 2015). Given the amount of data available and the current training-test design, the relative changes in the precipitation quantiles on daily time scale would have been derived by comparing empirical cumulative distribution functions (ECDFs) based on 30 and 19 values, respectively. This may yield unrealistic estimates for individual days and require additional steps of processing. A pooling procedure is thus necessary to obtain robust and stable estimates of the changes. In this study, the method was applied to pooled daily data separately for each calendar month, i.e. on the monthly scale. This is also a rather common procedure to take into account possible biases in the seasonal cycle (e.g. Räty et al., 2018; Meyer et al., 2019). Alternatively, one can apply a 30-day window centered on the day of interest. This might arguably be better in terms of continuity/smoother transition at the turn of the month. However, for most of the days, the differences are minimal. For computational efficiency, we opted for the monthly scale. We will include this information in Section 3.5.

(3) Sequencing of dry days (and wet days) is very important in many regions, hence if straight forward I would contemplate including a figure that captures this information.

**Reply**: Thank you for the comment. We found that the distribution of dry spells is similar across all simulations for a given catchment, where the majority consist of the short-term cases and a drought event longer than two weeks is rare. In Figure 4 below, it appears all simulations are more successful in modeling the dry spells at the larger catchment Gaulfoss than the smaller coastal catchment Oeyungen. However, the difference associated with the size of catchment can not be generalized due to the limited number of catchments in the study area. We will add this in the results section. We have chosen not to show the figures in the paper. Nonetheless, we are happy to reconsider this if needed.

(4) It would be good to have a climatology map and a DEM for the catchment shown in Figure 3, it would help with interpretation of bias pattern.

**Reply**: We will include such information in the results section, see Figure 5 below.

**Minor comments**

P1. Line 17. What do you mean by 'changed runoff'' (magnitude, seasonal flow, low/high flow metrics?). Perhaps qualify this a little, otherwise it looks a little 'hand-wavy'.

**Reply**: This will be corrected to "changed runoff magnitude".

[Figure]

Figure 4: Distribution of dry spells for the seNorge data in the training period 1957-1986 and the test period 1987-2005, as well as for six simulations of the test period. The results are aggregated over all grid cells in the catchments Gaulfoss (top) and Oeyungen (bottom).

[Figure]

Figure 5: Annual Precipitation (left) in the period 1957-2005 and elevation map (right) of the catchment Gaulfoss.

P2. Line 24. RCMs can explicitly resolve some process, though at 10-15 km, many processes relevant to rainfall events are parameterised. It would be more relevant here to speak about the convective permitting regional climate models that operate on 1- 4 km resolution (the recent UK national projections provide output on grid resolution just over 2km). The RCM modelling community is now pushing more towards the CPM scale rather than the 10-15 km scale. There are many recent overview type publications that you could cite here e.g:

Prein, A.F., Rasmussen, R., Castro, C.L. et al. Special issue: Advances in convection-permitting climate modeling. Clim Dyn 55, 1–2 (2020). https://doi.org/10.1007/s00382-020-05240-3

**Reply**: Thank you for bringing these recent publications to our attention. We will cite the mentioned paper in the revised manuscript.

P3. Line 29-30. This sentence has a few errors in it, 'which' should be 'with' and 'differ' should be 'different'.

**Reply**: We have rewritten this sentence to read: "For an accurate assessment of climate impact, one goal is thus to generate high-resolution realizations of future climate with the same distributional properties as the historical data product, except for potential changes in these distributional properties due to climate change."

P4 Line 87 – whilst I am not too familiar with Norwegian climatology, I would suggest that Norway as a whole is maritime, compared to continental climates.

**Reply**: Since Norway has a complex topography and the seNorge data we use has very fine spatial resolution, we carefully consider that the nine catchments belong to two climatic groups: oceanic/maritime and continental, according to the modified Köppen-Geiger climate classification method applied by Yang et al. (2020).

L403-408. To the reader this is a little confusing:

"Looking at the linear trend coefficient in the probit model, it seems that the seNorge data upscaled to 12 km resolution are generally able to capture the change that there are proportionally more wet days in the test period than in the training period, while the RCM

data either project strong negative changes or simply no change in most catchments. For the gamma model, both RCMs seem to have projected correct changes in the trend and seasonality. For this reason, it cannot be expected that the RCM runs perfectly reproduce the precipitation patterns in the finer-scale seNorge data"

The penultimate sentence in this section says " For the gamma model, both RCMs seem to have projected correct changes in the trend and seasonality" which is a positive aspect of the model. Therefore, when reading the last sentence "For this reason, it cannot be expected that the RCM runs perfectly reproduce the precipitation patterns in the finer-scale seNorge data" one is a little surprised. Presumably the performance issues relate to the first sentence in this section? To make this a little more understandable, I would replace "For this reason" in your last sentence with the actual reason, presumably the somewhat different trend coefficient in the RCMs? It is also worth noting that RCMs are not hind-casts (unless you were using reanalysis forced RCMs), rather largely free running models (following the global response to observed emissions as simulated by the driving GCM), hence you could easily end up with somewhat different temporal trends, particularly on such short time frames.

**Reply**: Thank you for pointing out this contradictory messaging. We have rewritten the overall statement to read:

"The historical RCM runs used here are available until 2005, and the observation-based data is available from 1957, yielding a data set with 49 years of data. With 30 years of data used to train the models, this leaves only 19 years of data for the out-of-sample evaluation. With only 19 years of data in the test period, we may expect to see some effects of natural variability when comparing the seNorge data product and the largely free running RCMs. Looking at the linear trend coefficient in the probit model, it seems that the seNorge data upscaled to 12 km resolution are generally able to capture the change that there are proportionally more wet days in the test period than in the training period, while the RCM data either project strong negative changes or simply no change in most catchments. For the gamma model, however, both RCMs seem to have projected correct changes in the trend and seasonality. Overall, we see that all versions of the WG method yield better performance than the marginal persistence reference method based on seNorge data from 1957-1986, and including RCM information improves upon the stationary WG approach."

**References**

Andreoli, R. V. and Kayano, M. T.: ENSO-related rainfall anomalies in South America and associated circulation features during warm and cold Pacific decadal oscillation regimes, International Journal of Climatology, 25, 2017–2030, URL https://doi.org/10.1002/joc.1222, 2005.

Cannon, A. J., Sobie, S. R., and Murdock, T. Q.: Bias Correction of GCM Precipitation by Quantile Mapping: How Well Do Methods Preserve Changes in Quantiles and Extremes?, Journal of Climate, 28, 6938–6959, URL https://doi.org/10.1175/jcli-d-14-00754.1, 2015.

Meyer, J., Kohn, I., Stahl, K., Hakala, K., Seibert, J., and Cannon, A. J.: Effects of univariate and multivariate bias correction on hydrological impact projections in alpine catchments, Hydrology and Earth System Sciences, 23, 1339–1354, URL https://doi.org/10.5194/hess-23-1339-2019, 2019.

Räty, O., Räisänen, J., Bosshard, T., and Donnelly, C.: Intercomparison of Univariate and Joint Bias Correction Methods in Changing Climate From a Hydrological Perspective, Climate, 6, 33, URL https://doi.org/10.3390/cli6020033, 2018.

Wu, R., Hu, Z.-Z., and Kirtman, B. P.: Evolution of ENSO-related rainfall anomalies in East Asia, Journal of Climate, 16, 3742–3758, 2003.

Yang, X., Magnusson, J., Huang, S., Beldring, S., and Xu, C.-Y.: Dependence of regionalization methods on the complexity of hydrological models in multiple climatic regions, Journal of Hydrology, 582, 124357, URL https://doi.org/10.1016/j.jhydrol.2019.124357, 2020.

---

## Author Comment (AC2)

**Bridging the scale gap: Obtaining high-resolution stochastic simulations of gridded daily precipitation in a future climate**

Qifen Yuan, Thordis L. Thorarinsdottir, Stein Beldring,
Wai Kwok Wong, Chong-Yu Xu

**Response to Referee 2**

We thank the reviewer for his/her positive comments on the manuscript. Below is our response to the issues raised in the review.

**General comments**

This manuscript is very interesting for the topic of rainfall fields at hydrological scales.

My comments only regard suggestions of minor revisions, in order to slightly improve the quality of this interesting manuscript.

**Reply**: Thank you for your positive evaluation of our work. We will modify the manuscript following your suggestions.

**Comment 1**

In the introduction, Authors should enrich the state-of-the-art of stochastic models, by mentioning Neymann-Scott and Bartlett-Lewis families, also available for transient versions (Burton et al., 2008, 2010; Cowpertwait et al., 2002; De Luca et al., 2020)

References:

Burton, C.G. Kilsby, H.J. Fowler, P.S.P. Cowpertwait, P.E. O'Connell, RainSim: A spatial–temporal stochastic rainfall modelling system, Environmental Modelling & Software, Volume 23, Issue 12, 2008 https://www.sciencedirect.com/science/article/abs/pii/S1364815208000613

Burton, A., H. J. Fowler, C. G. Kilsby, and P. E. O'Connell (2010), A stochastic model for the spatial–temporal simulation of nonhomogeneous rainfall occurrence and amounts, Water Resour. Res., 46, W11501, doi:10.1029/2009WR008884

Cowpertwait, P. S. P., Kilsby, C. G., and O'Connell, P. E., A space-time Neyman-Scott model of rainfall: Empirical analysis of extremes, Water Resour. Res., 38( 8), doi:10.1029/2001WR000709, 2002.

De Luca, D.L.; Petroselli, A.; Galasso, L. (2020). A Transient Stochastic Rainfall Generator for Climate Changes Analysis at Hydrological Scales in Central Italy. Atmosphere, 11(12), 1292. https://doi.org/10.3390/atmos11121292 (https://www.mdpi.com/2073-4433/11/12/1292)

**Reply**: Thank you for your suggestion. We will add a paragraph in the introduction to give a proper review of these state-of-the-art stochastic models.

**Comment 2**

In Section 3, I suggest to insert a flow chart in order to make clearer for a reader the several steps of the proposed procedure.

**Reply**: This is indeed a very good point. We will add a conceptual graphic, see Figure 1 below, to outline the different components of the proposed procedure.

[Figure]

Figure 1: The proposed two-stage weather generator approach for simulations of fine-scale daily precipitation in a future climate.

---

## Author Response (AR1)

**Bridging the scale gap: Obtaining high-resolution stochastic simulations of gridded daily precipitation in a future climate**

Qifen Yuan, Thordis L. Thorarinsdottir, Stein Beldring,
Wai Kwok Wong, Chong-Yu Xu

We thank the editors for handling the manuscript, and are grateful to two anonymous reviewers for their insightful comments. We have done our best to revise the paper according to their suggestions. Point-by-point responses are given below. Please note that the line numbers in reviewer's comments refer to the original manuscript "hess-2020-673-manuscript-version1.pdf", while in our reply refer to the revised version.

**Response to Referee 1**

This is a very meaningful study that puts forward a combination of statistical methods and approaches to produce fine scale resolution rainfall estimates for catchments (small to medium/large-ish sizes). It combines aspects of the weather-generator type approach with spatial statistics, acknowledging the need to also consider temporal dependencies and structures.

The language and graphics are clear and concise, explicit formulas are provided, accompanied by relevant references to underpinning methods and reference to used R-packages/methods.

Whilst I think that there are a one or two of issues to address method wise before it can be applied in a climate change context, I think it is a meaningful suggestion to build on. Overall, in my opinion this manuscript offers a very meaningful contribution to the peer-literature on statistical downscaling and merits publication with only minor edits.

**Reply**: Thank you for your positive evaluation of our work. We have modified the manuscript following your suggestions. Please find details below.

**General comments**

Given the focus of this paper, I suggest a few changes that would make the paper more accessible to a wider audience that should pay attention to this work but would find it

difficult to detangle the methodology (the paper has a very strong statistical flavour and assumes that readers are familiar with statistical terminology). *Firstly*, I suggest that you put in a conceptual graphic, that outlines the different components in our analysis and how they fit together, this will help the reader understand how the different model parts fits together, and where the 'change' is applied (change between the two time periods). *Second*, I think it will help the reader if you outline when you expect the model to work well, and when you expect there to be difficulties in the methods section. Some of the performance issues discussed in the discussion section are entirely predictable and I think it will help the reader understand the method better if you have a paragraph that speaks about the expected strengths and expected weaknesses before you conduct the experiments, this will get the reader introduced to how the method works, and then the results will make more sense. I think the manuscript will be more accessible and thus more successful if it embraces a wider audience.

**Reply**: Firstly, we have included a conceptual graphic in the introduction section, see Figure 1 in the revised manuscript. Second, we have added a paragraph in the introduction section (lines 61-69) where we state the two main strengths of our method as well as its two main assumptions.

One aspect not discussed in this paper is the applicability in a climate change context. Whilst the two time periods do exhibit some climatological differences, changes that one might expect in 20-50 years time are likely to be more pronounced. In this context, you may need to include a representation of oscillatory behaviour in the spatial dependency structure (in addition to a seasonal representation). This may not be the case for Norway, but other regions with more variable rainfall climatologies (strongly influenced by e.g. ENSO or IDO ... or any other large-scale climate oscillation or teleconnection pattern. I think it would be very meaningful if you could elaborate on this need in your discussion section, and perhaps propose a couple of tests to see if this is necessary, e.g. compare spatial correlation in sub-sets of RCM rainfall fields that capture certain seasonal/oscillation modes – are these still similar enough (how to judge?) to what we see in the coarse resolution observed rainfall fields?

**Reply**: This is indeed a very important discussion point for a practical application of our method in climate change impact studies. We have added a paragraph to the discussion section (lines 433-441).

**Four smaller issues**

(1) I think it is always useful to be critical of the RCM – hence I think the authors should point out the need to demonstrate that the RCMs selected for the analysis do indeed capture the spatial and temporal patterns of the variable of interest on its resolution before down-scaling (where the downscaling is drawing on a relationship derived from observed data).

**Reply**: It is useful to check whether the spatial and temporal patterns of the RCM outputs are realistic at its own scale. In the newly added paragraph in the introduction, we have pointed out that for the success of the method the climate change signal projected by the RCMs must be correct and transferrable between the two spatial scales (lines 66-69).

(2) Using empirical scaling (or decile scaling) as a reference is meaningful because that is a very common method. However, many water-related studies would use daily scaling rather than monthly scaling as you are draping a scaling coefficient of monthly resolution onto a variable with daily resolution, then evaluating on the daily resolution. The comparison seems a little unfair. I see no need to redo your analysis, but I would point out how the monthly scaling could impact the comparison.

**Reply**: We have now briefly discussed this in Section 3.5 (lines 275-278).

(3) Sequencing of dry days (and wet days) is very important in many regions, hence if straight forward I would contemplate including a figure that captures this information.

**Reply**: In the results section (lines 395-398) we have added that we found the distribution of dry spells is similar across different methods for any given catchment in our case study. We have chosen not to show the figures. Nonetheless, we are happy to reconsider this if needed.

(4) It would be good to have a climatology map and a DEM for the catchment shown in Figure 3, it would help with interpretation of bias pattern.

**Reply**: We have added this in the results section, see Figure 5 in the revised manuscript.

**Minor comments**

P1. Line 17. What do you mean by 'changed runoff'' (magnitude, seasonal flow, low/high flow metrics?). Perhaps qualify this a little, otherwise it looks a little 'hand-wavy'.

**Reply**: This has been corrected, see line 17.

P2. Line 24. RCMs can explicitly resolve some process, though at 10-15 km, many processes relevant to rainfall events are parameterised. It would be more relevant here to speak about the convective permitting regional climate models that operate on 1- 4 km resolution (the recent UK national projections provide output on grid resolution just over 2km). The RCM modelling community is now pushing more towards the CPM scale rather than the 10-15 km scale. There are many recent overview type publications that you could cite here e.g:

Prein, A.F., Rasmussen, R., Castro, C.L. et al. Special issue: Advances in convection-permitting climate modeling. Clim Dyn 55, 1–2 (2020). https://doi.org/10.1007/s00382-020-05240-3

**Reply**: Thank you for bringing these recent publications to our attention. We have cited the mentioned paper in the revised manuscript (line 26) .

P3. Line 29-30. This sentence has a few errors in it, 'which' should be 'with' and 'differ' should be 'different'.

**Reply**: We have rewritten this sentence, see lines 30-32.

P4 Line 87 – whilst I am not too familiar with Norwegian climatology, I would suggest that Norway as a whole is maritime, compared to continental climates.

**Reply**: Since Norway has a complex topography and the seNorge data we use has very fine spatial resolution, we carefully consider that the nine catchments belong to two climatic groups: oceanic/maritime and continental, according to the modified Köppen-Geiger climate classification method applied by Yang et al. (2020). Therefore, we have not made any change to the text which now appears in lines 100-101 in the revised manuscript.

L403-408. To the reader this is a little confusing:

"Looking at the linear trend coefficient in the probit model, it seems that the seNorge data upscaled to 12 km resolution are generally able to capture the change that there are proportionally more wet days in the test period than in the training period, while the RCM data either project strong negative changes or simply no change in most catchments. For the gamma model, both RCMs seem to have projected correct changes in the trend and seasonality. For this reason, it cannot be expected that the RCM runs perfectly reproduce the precipitation patterns in the finer-scale seNorge data"

The penultimate sentence in this section says " For the gamma model, both RCMs seem to have projected correct changes in the trend and seasonality" which is a positive aspect of the model. Therefore, when reading the last sentence "For this reason, it cannot be expected that the RCM runs perfectly reproduce the precipitation patterns in the finer-scale seNorge data" one is a little surprised. Presumably the performance issues relate to the first sentence in this section? To make this a little more understandable, I would replace "For this reason" in your last sentence with the actual reason, presumably the somewhat different trend coefficient in the RCMs? It is also worth noting that RCMs are not hind-casts (unless you were using reanalysis forced RCMs), rather largely free running models (following the global response to observed emissions as simulated by the driving GCM), hence you could easily end up with somewhat different temporal trends, particularly on such short time frames.

**Reply**: Thank you for pointing out this contradictory messaging. We have rewritten the overall statement, see lines 422-431.

**Response to Referee 2**

**General comments**

This manuscript is very interesting for the topic of rainfall fields at hydrological scales.

My comments only regard suggestions of minor revisions, in order to slightly improve the quality of this interesting manuscript.

**Reply**: Thank you for your positive evaluation of our work. We have modified the manuscript following your suggestions. Please find details below.

**Comment 1**

In the introduction, Authors should enrich the state-of-the-art of stochastic models, by mentioning Neymann-Scott and Bartlett-Lewis families, also available for transient versions (Burton et al., 2008, 2010; Cowpertwait et al., 2002; De Luca et al., 2020)

References:

Burton, C.G. Kilsby, H.J. Fowler, P.S.P. Cowpertwait, P.E. O'Connell, RainSim: A spatial–temporal stochastic rainfall modelling system, Environmental Modelling & Software, Volume 23, Issue 12, 2008 https://www.sciencedirect.com/science/article/abs/pii/S1364815208000613

Burton, A., H. J. Fowler, C. G. Kilsby, and P. E. O'Connell (2010), A stochastic model for the spatial–temporal simulation of nonhomogeneous rainfall occurrence and amounts, Water Resour. Res., 46, W11501, doi:10.1029/2009WR008884

Cowpertwait, P. S. P., Kilsby, C. G., and O'Connell, P. E., A space-time Neyman-Scott model of rainfall: Empirical analysis of extremes, Water Resour. Res., 38( 8), doi:10.1029/2001WR000709, 2002.

De Luca, D.L.; Petroselli, A.; Galasso, L. (2020). A Transient Stochastic Rainfall Generator for Climate Changes Analysis at Hydrological Scales in Central Italy. Atmosphere, 11(12), 1292. https://doi.org/10.3390/atmos11121292 (https://www.mdpi.com/2073-4433/11/12/1292)

**Reply**: Thank you for your suggestion. We have given a proper review of these state-of-the-art stochastic models in the introduction, see lines 35-39.

**Comment 2**

In Section 3, I suggest to insert a flow chart in order to make clearer for a reader the several steps of the proposed procedure.

**Reply**: We have added a conceptual graphic Figure 1 in the introduction, and using that as a guideline we introduce the main steps of the proposed procedure to our readers already in the introduction, see lines 61-69.

**References**

Yang, X., Magnusson, J., Huang, S., Beldring, S., and Xu, C.-Y.: Dependence of regionalization methods on the complexity of hydrological models in multiple climatic regions, Journal of Hydrology, 582, 124357, URL `https://doi.org/10.1016/j.jhydrol.2019.124357`, 2020.